# Diagnostic and health service pathways to diagnosis of cancer-registry notified cancer of unknown primary site (CUP)

**Andrea L. Schaffer** [1] *, **Sallie-Anne Pearson**[1], **Oscar Perez-Concha**[1], **Timothy Dobbins**[2], **Robyn L. Ward**[3], **Marina T. van Leeuwen**[1], **Joel J. Rhee**[4,5], **Maarit A. Laaksonen**[1], **Glynis Craigen**[6], **Claire M. Vajdic**[1]

**1** Centre for Big Data Research in Health, UNSW Sydney, Sydney, Australia, **2** National Drug and Alcohol Research Centre, UNSW Sydney, Sydney, Australia, **3** Faculty of Medicine and Health, University of Sydney, Sydney, Australia, **4** Centre for Primary Health Care and Equity, UNSW Sydney, Sydney, Australia, **5** School of Medicine, University of Wollongong, Wollongong, Australia, **6** Cancer Voices NSW, Sydney, Australia

* andrea.schaffer@unsw.edu.au

**Data Availability Statement:** The data used for this study are from the 45 and Up Study, which is an

## Abstract

### Background

Cancer of unknown primary (CUP) is a late-stage malignancy with poor prognosis, but we know little about what diagnostic tests and procedures people with CUP receive prior to diagnosis. The purpose of this study was to determine how health service utilisation prior to diagnosis for people with cancer-registry notified CUP differs from those notified with metastatic cancer of known primary.

### Methods

We identified people with a cancer registry notification of CUP (n = 327) from the 45 and Up Study, a prospective cohort of 266,724 people ≥45 years in New South Wales, Australia, matched with up to three controls with a diagnosis of metastatic cancer of known primary (n = 977). Baseline questionnaire data were linked to population health data to identify all health service use, diagnostic tests, and procedures in the month of diagnosis and 3 months prior. We used conditional logistic regression to estimate adjusted odds ratios (ORs) and 95% confidence intervals (CIs).

### Results

After adjusting for age and educational attainment, people with a cancer-registry notified CUP diagnosis were more likely to be an aged care resident (OR = 2.78, 95%CI 1.37–5.63), have an emergency department visit (OR = 1.65, 95%CI 1.23–2.21), serum tumor marker tests (OR = 1.51, 95%CI 1.12–2.04), or a cytology test without immunohistochemistry (OR = 2.01, 95%CI 1.47–2.76), and less likely to have a histopathology test without immunohistochemistry (OR = 0.43, 95%CI 0.31–0.59). Neither general practitioner, specialist, allied health practitioner or nurse consultations, hospitalisations, nor imaging procedures were associated with a CUP diagnosis.

open research resource managed by the Sax Institute (https://www.saxinstitute.org.au/our-work/45-up-study/for-researchers/). The datasets for this study were created by linkage of the 45 and Up Study baseline survey data to Australian Government and NSW state data sources with support from the NSW Centre for Health Record Linkage (www.cherel.org.au), and permission from the custodians of the datasets under specific ethics approvals for pre-specified analyses only. These data contain sensitive information and are potentially re-identifiable, and we do not have permission to share them with third parties. The data used for this study are available to researchers pending appropriate ethics and data access requirements. Interested researchers can contact the Sax Institute (45andup.research@saxinstitute.org.au) and NSW Centre for Health Record Linkage (cherel.mail@moh.health.nsw.gov.au) for data access approval procedures.

**Funding:** This work was supported by a Cancer Institute of New South Wales (NSW) Epidemiology Linkage Program Grant (10/EPI/2-06). AS was supported by a National Health and Medical Research Council (NHMRC) Early Career Fellowship (#1158763). The funders had no role in the study design, data collection and analysis, decision to publish, or preparation of the manuscript.

**Competing interests:** The authors have declared that no competing interests exist.

**Abbreviations:** CI, confidence interval; CT, computerized tomography; CUP, cancer of unknown primary; ED, emergency department; GP, general practitioner; IQR, interquartile range; MBS, Medicare Benefits Scheme; MRI, magnetic resonance imaging; NSW, New South Wales; OR, odds ratio; PBS, Pharmaceutical Benefits Scheme; SPECT, single photon emission computerized tomography.

## Conclusions

The health service and diagnostic pathway to diagnosis differs markedly for people notified with CUP compared to those with metastatic cancer of known primary. While these differences may indicate missed opportunities for earlier detection and appropriate management, for some patients they may be clinically appropriate.

## Introduction

Cancer of unknown primary site (CUP) is the 14th most commonly diagnosed cancer and the 6th most common cause of cancer death in Australia.[1] It is characterized by metastatic disease with an unidentified primary site, and extremely poor survival;[2–5] in Australia, the 5-year survival rate is 13%.[6] Population-based CUP incidence rates have declined over time, [2,6,7] a trend that can most likely be attributed to advances in diagnostic investigations. While a diagnosis of CUP ideally should be limited to people with a histological confirmation of metastatic cancer, in whom thorough testing has failed to identify the site of the primary tumor, people registered with a diagnosis of CUP in population-based cancer registries are heterogeneous, with many people receiving a diagnosis based on clinical examination only. [4,5,8–11] Thus, it is difficult to compare population-based studies of registry-notified CUP to cases series of "true CUP" cases, who have received exhaustive investigations.

Studies of how people diagnosed with CUP interact with the health system prior to their definitive diagnosis, and what investigations they receive, are limited.[8,12] Although there are no Australian CUP clinical practice guidelines, the US National Comprehensive Cancer Network (NCCN) and the European Society for Medical Oncology (ESMO) CUP-specific guidelines support the judicious use of diagnostic investigations, in keeping with a patient's prognosis and treatment options.[13,14] Yet, the diagnostic pathway of patients with a cancer-registry notified CUP diagnosis is highly variable,[8,15,16] which may reflect heterogeneity in clinical presentation and patient characteristics.

Previous studies suggest that the diagnostic pathway for people with CUP differs compared to people diagnosed with other cancers, including late-stage cancers of known primary site.[8] CUP appears more likely to be diagnosed following an emergency department visit compared with all other cancers, and diagnosis may involve less specialist input and fewer invasive diagnostic procedures, including histopathological diagnosis of cancer.[8,15] In this prospective Australian cohort study, we sought to compare the pathway to diagnosis for people diagnosed with a cancer-registry CUP diagnosis, compared with people diagnosed with metastatic cancer of known primary site.

## Materials and methods

### Data sources

The Sax Institute's 45 and Up Study[17] is a prospective cohort study with comprehensive information on self-reported lifestyle behaviors and a range of health, functional and social measures at baseline. New South Wales (NSW) residents aged at least 45 years were randomly sampled from the Department of Human Services (formerly Medicare Australia) enrollment database, which provides near complete coverage of the population. People 80+ years of age and residents of rural and remote areas were oversampled. A total of 266 933 individuals joined the study by completing a postal questionnaire between 2006 and 2009. The date of completion of the questionnaire was considered the baseline period. Around 18% of those

invited participated, and the cohort included 11% of the NSW population aged 45 years or more.

Australia's publicly funded health care system provides all citizens and permanent residents with a range of health services including treatment in public hospitals, subsidized treatment in private hospitals, subsidized outpatient services including consultations, procedures and tests, and subsidized medicines. Records of these transactions are made available for ethically approved health research. The 45 and Up Study cohort was probabilistically linked to population-based administrative health datasets by the Centre for Health Record Linkage to identify incident cancers, comorbid health conditions, subsidized health services and deaths. The datasets were: (i) the NSW Cancer Registry, a population-based registry of incident invasive cancer diagnoses (excluding basal and squamous cell carcinoma of the skin) in NSW 1994–2012; (ii) the NSW Admitted Patients Data Collection 2001–2015; (iii) the NSW Emergency Department Data Collection 2005–2016; (iv) the Medicare Benefits Scheme (MBS) 2001–2015; (v) the Pharmaceutical Benefits Scheme (PBS) 2004–2015; and (vi) the NSW Registry of Births, Deaths and Marriages 2006–2016. We excluded 209 cohort participants because they did not have a linked MBS or PBS record.

## Study population

As previously described, we defined cases as cohort participants with a cancer registry diagnosis of CUP (ICD-10-AM codes C80, C76, C26 or C39) more than 3 months after baseline. [18,19] We included all people with a registry diagnosis of CUP, regardless of whether it was histopathologically confirmed. We randomly selected a control group with a cancer registry diagnosis of solid metastatic cancer of known primary site; for the controls, the first manifestation of this cancer was metastatic disease, with a recorded extent of disease spread that was either distant or regional, on the basis of the highest degree of spread (extent of spread of cancer from its point of origin) classified by the NSW Cancer Registry. Individuals with another registered cancer diagnosis in the same month were also eligible for inclusion as a case or control.

We matched the controls to cases by month and year of completion of the baseline questionnaire in the cohort and by month and year of cancer diagnosis. We selected up to three controls per case using incidence density sampling with replacement.[20] We did not match on age and sex as we wished to examine the effect of these factors.

## Health service use

We quantified health services accessed in the month of diagnosis and the three months prior to diagnosis, including hospitalisations, emergency department (ED) visits, and consultations with general practitioners (GP), specialists, allied health practitioners, and nurses, and residence in an aged care facility. As the NSW Cancer Registry only provides the month and year of cancer diagnosis, we ascertained all health services in the entire month of diagnosis. We focused on the month of diagnosis and the three months prior to diagnosis based on our previous work showing that health service utilisation increased around this time [8], and clinical advice that this was the most relevant time period for diagnostic investigation. We confirmed that these observations held true for our cohort (data not shown).

We also identified the subset of GP consultations in the home, an institution or hospital (MBS item numbers 4, 24, 37, 47, 58, 59, 60, 65, 5003, 5023, 5043, 5063, 5220, 5223, 5227, 5228) as a marker of patient frailty; we also distinguished GP consultations that occurred in the doctors' rooms (doctor's offices) (MBS item numbers 2, 3, 23, 35, 36, 44, 52, 53, 54, 57, 2501, 2504, 2517, 2521, 2525, 2546, 2552, 2721, 2725, 5000, 5020, 5040, 5060). We further

distinguished GP consultations for the preparation, contribution or review of a GP management plan or multidisciplinary/team care plan (MBS item numbers 721, 723, 729, 731, 732, 735, 739, 743, 747, 750, 758, 820, 822, 823, 825, 826, 828, 830, 832, 834, 835, 837, 838, 900). We separately identified specialist consultations for the initial assessment or review of patients with at least two comorbidities, classified as complex cases (MBS item numbers 132, 133). All health service use variables were dichotomized (any vs none).

## Diagnostic tests and procedures

We ascertained all cancer-related imaging, endoscopy, medical procedures and pathology tests performed in tertiary and community settings during the month of diagnosis and the three months prior. We identified X-rays, computerized tomography (CT) scans, single photon emission computerized tomography (SPECT), ultrasound, magnetic resonance imaging (MRI), nuclear imaging and endoscopy. We classified medical procedures as exploratory surgery, non-surgical resection (such as fine needle aspiration, biopsy), and surgical resection (both non-cutaneous and cutaneous). The pathology tests of interest were one or more serum tumor markers (beta-2 microglobulin, alpha-fetoprotein, ca-15.3 antigen, ca-125 antigen, ca-19.9 antigen, cancer associated serum antigen, carcinoembryonic antigen, human chorionic gonadotrophin, neuron specific enolase, thyroglobulin, and prostate specific antigen; MBS item numbers 66629, 66650, 66651, 66652, 66653, 66655), cytology and histopathology alone or with immunostaining (immunocytochemistry or immunohistochemistry), and cytogenetics. These tests are most commonly used to help identify the tissue of origin for CUP.[14,21]

## Statistical analysis

We used conditional logistic regression to estimate the odds of a cancer-registry notified CUP diagnosis associated with health service use and diagnostic tests and procedures. We first modelled each factor individually adjusted by age and sex only, and those variables with $p < 0.20$ were considered for inclusion in the fully adjusted multivariable model. We used the 45 and Up Study baseline questionnaire to identify potential confounding factors, such as self-reported overall health and thus fitness for diagnostic investigation. To avoid multicollinearity between similar measures, we assessed the association between pairs of factors using Cramér's V statistic and those with a correlation coefficient $\geq 0.25$ were considered correlated. This approach was also used to identify tests and procedures that were on the pathway between consultations and visits and a cancer diagnosis, to avoid over-adjustment.

We built conditional logistic regression models using backward elimination, stopping when the remaining variables in the model were all significantly associated with CUP ($p < 0.05$). We built multiple multivariable models with all possible combinations of non-correlated variables, and then selected the model with the lowest Akaike Information Criterion as the final model.

## Ethics

The study was approved by the NSW Population and Health Services and Human Research Ethics Committee (2012/11/428) and the 45 and Up Study was approved by the University of New South Wales Human Research Ethics Committee (HREC 15408). All participants provided written informed consent at the time of recruitment for follow-up and linkage of their information to administrative health databases. All procedures were in accordance with the ethical standards of the ethics committees mentioned above and with the 1964 Helsinki declaration and its later amendments.

## Results

We identified 327 incident cases of CUP and 977 matched incident solid metastatic cancer controls, diagnosed between 2006 and 2012. The median age at diagnosis of CUP was 76 years (interquartile range, IQR: 66–82 years) and 68 years (IQR: 60–76 years) for solid metastatic cancer controls. The median time from completion of the baseline questionnaire to cancer diagnosis was 33 months (IQR: 21–46 months). Of the people with a registry-notified CUP diagnosis, 165 (50.5%) had a histopathogical diagnosis, 49 (15.0%) a cytological diagnosis (including fine needle aspiration, smears, washing and sputum), 88 (26.9%) a clinical diagnosis (including clinical, imaging and biochemical procedures) and 25 (7.6%) were identified by death certificate only. Of the CUP cases (n = 165) registered with a NSW Cancer Registry histopathological diagnosis, 20 (12.1%) tumors were carcinoma (subtype not specified), 81 (49.1%) were carcinoma (subtype specified), 50 (30.3%) were adenocarcinoma, and 14 (8.5%) were other morphological types. Of the 870 people with metastatic cancer of known primary with a histopathological diagnosis, 407 (46.8%) tumors were carcinoma, 395 (45.4%) were adenocarcinoma, and 68 (7.8%) were other. The most common primary sites for the solid metastatic cancer controls were breast (C50; n = 168), bronchus and lung (C34; n = 163), colon (C18; n = 152), prostate (C61; n = 123) and rectum (C20; n = 57).

In the month of diagnosis and the three months prior, people with a cancer-registry notified CUP diagnosis were more likely to have an ED visit (odds ratio, OR = 1.84, 95% confidence interval, CI 1.41–2.40) and less likely to have a specialist consultation (OR = 0.66, 95%CI 0.48–0.92), compared to solid metastatic cancer controls (Table 1) in models adjusted for age and sex only. During this period there was no difference between the two metastatic cancer groups in terms of hospitalisation or consultations with a GP, allied health practitioner, nurse, or specialist indicating complex care. People diagnosed with CUP were more than 3-fold more likely

**Table 1. Association between recent health service use and diagnosis of CUP compared to metastatic cancer of known primary.**

| Health service use in three months prior and month of diagnosis | CUP (n = 327) | Metastatic cancer, known primary site (n = 977) | Age- and sex-adjusted OR (95% CI) |
|---|---|---|---|
| | n (%) | n (%) | |
| **Tertiary care** | | | |
| Hospitalisation | 221 (67.6) | 682 (69.8) | 0.88 (0.66–1.17) |
| Emergency department visit | 172 (52.6) | 316 (32.3) | 1.84 (1.41–2.40) |
| **Consultations** | | | |
| General practitioner (GP) | 305 (93.3) | 932 (95.3) | 0.79 (0.44–1.42) |
| Consulting room (including after-hours visits)[a] | 297 (90.8) | 920 (94.2) | 0.78 (0.47–1.29) |
| Home, institution or hospital (including after-hours visits)[b] | 49 (15.0) | 79 (8.1) | 1.43 (0.96–2.13) |
| Management and multidisciplinary care plans | 64 (19.6) | 143 (14.6) | 1.27 (0.91–1.78) |
| Specialist/consultant physician | 244 (74.6) | 825 (84.4) | 0.66 (0.48–0.92) |
| Complex case | 51 (15.6) | 121 (12.4) | 1.30 (0.89–1.90) |
| Allied health practitioner | 82 (25.1) | 245 (25.1) | 0.96 (0.71–1.31) |
| Nurse | 39 (11.9) | 127 (13.0) | 0.84 (0.56–1.26) |
| **Residence in aged care facility[c]** | 32 (9.8) | 19 (1.9) | 3.42 (1.78–6.56) |

[a] MBS item numbers 2, 3, 23, 35, 36, 44, 52, 53, 54, 57, 2501, 2504, 2517, 2521, 2525, 2546, 2552, 2721, 2725, 5000, 5020, 5040, 5060.

[b] MBS item numbers 4, 24, 37, 47, 58, 59, 60, 65, 5003, 5023, 5043, 5063, 5220, 5223, 5227, 5228.

[c] EDDC referral source = 5 (Residential Aged Care facility); APDC: source of referral 6 (Nursing home/Residential Aged Care Facility) or mode of separation = 3 (transferred to nursing home) or peer group = F2 (Nursing home) and MBS item numbers: 20, 35, 43, 51, 92, 93, 95, 96, 5010, 5028, 5049, 5067, 5260, 5263, 5265, 5267, 2125, 2138, 2179, 2220, 82223, 82224, 82225, 73934, 73935, 10984, 903, 731.

to be in aged care (OR = 3.42, 95%CI 1.78–6.56). We found people diagnosed with CUP were less likely than controls to have an endoscopy, a surgical (non-cutaneous) resection, and histopathology (Table 2). Conversely, they were more likely to have serum tumor marker tests and cytology. Overall, 21 (6.4%) people diagnosed with CUP did not have any cancer-related investigations, as defined by imaging procedures, endoscopy, medical procedures (e.g. resection), or pathology tests. Among controls, 60 (6.1%) did not have any cancer-related investigations.

In our fully adjusted model, the only confounding factors were age and educational attainment, with those diagnosed with CUP more likely to be older and to have no school certificate (less than 4 years of secondary education) (Table 3).[18] After adjustment for these factors, and the mutual adjustment for health service use, diagnostic tests and procedures, the only variables that remained associated with an increased probability of a CUP diagnosis were being in an aged care facility (OR = 2.78, 95%CI 1.37–5.63), ≥1 ED visit (OR = 1.65, 95%CI 1.23–2.21), and the following pathology tests: serum tumor marker tests (OR = 1.51, 95%CI 1.12–2.04), and cytology without immunohistochemistry (OR = 2.01, 95%CI 1.47–2.76). People diagnosed with CUP were less than half as likely to have had a histopathology test without immunohistochemistry (OR = 0.43, 95%CI 0.31–0.59).

## Discussion

In a contemporary cohort of Australian adults, we observed differences in the pathways to diagnosis for people with a cancer-registry notified CUP diagnosis compared with people

**Table 2. Association between recent cancer-related diagnostic tests and procedures and diagnosis of CUP compared to metastatic cancer of known primary.**

| Tests or procedures in month of diagnosis and three months prior | CUP (n = 327) | Metastatic cancer, known primary (n = 977) | Age- and sex-adjusted OR (95% CI) |
|---|---|---|---|
| | n (%) | n (%) | |
| **Imaging procedures** | | | |
| X-ray | 165 (50.5) | 493 (50.5) | 0.99 (0.76–1.29) |
| Computerized tomography (CT) | 213 (65.1) | 618 (63.3) | 1.17 (0.89–1.55) |
| Single photon emission computerized tomography (SPECT) | 49 (15.0) | 159 (16.3) | 1.17 (0.81–1.69) |
| Ultrasound | 141 (43.1) | 436 (44.6) | 1.24 (0.94–1.62) |
| Magnetic resonance imaging (MRI) | 14 (4.3) | 38 (3.9) | 1.37 (0.71–2.62) |
| Nuclear imaging | 50 (15.3) | 174 (17.8) | 1.07 (0.74–1.53) |
| **Endoscopy** | 53 (16.2) | 277 (28.4) | 0.55 (0.39–0.77) |
| **Medical procedures** | | | |
| Exploratory surgery[a] | c | c | 1.36 (0.21–8.71) |
| Resection, non-surgical[b] | 290 (88.7) | 882 (90.3) | 1.05 (0.69–1.60) |
| Resection, surgical (non-cutaneous) | 20 (6.1) | 158 (16.2) | 0.46 (0.28–0.75) |
| Resection, surgical (cutaneous) | 27 (8.3) | 68 (7.0) | 1.12 (0.68–1.85) |
| **Pathology tests** | | | |
| Serum tumor markers | 126 (38.6) | 327 (33.5) | 1.36 (1.04–1.78) |
| Cytology without immunohistochemistry | 112 (34.3) | 256 (26.2) | 1.78 (1.33–2.38) |
| Cytology with immunocytochemistry | 25 (7.6) | 36 (3.7) | 2.54 (1.45–4.46) |
| Histopathology without immunohistochemistry | 101 (30.9) | 561 (57.4) | 0.40 (0.29–0.53) |
| Histopathology with immunohistochemistry | 59 (18.0) | 252 (25.8) | 0.77 (0.55–1.08) |
| Cytogenetics | c | c | 5.11 (0.79–33.1) |

[a] Exploratory surgery includes laparotomy, thoracotomy, and cervical exploration of mediastinum

[b] Non-surgical resection includes fine needle aspiration, biopsy or excision, other than regional or radical excision which has been classified as surgical resection

[c] Cell size <5; exact cell size suppressed for privacy reasons

**Table 3. Association between recent health service use, cancer-related diagnostic tests and procedures, and diagnosis of CUP compared to metastatic cancer of known primary.**

| Tests or procedures in month of diagnosis and three months prior | Fully adjusted |
|---|---|
| | OR (95% CI)[a] |
| **Tertiary care** | |
| ≥1 emergency department visit | 1.65 (1.23–2.21) |
| **Aged care facility** | 2.78 (1.37–5.63) |
| **Pathology tests** | |
| Serum tumor markers | 1.51 (1.12–2.04) |
| Cytology without immunohistochemistry | 2.01 (1.47–2.76) |
| Histopathology without immunohistochemistry | 0.43 (0.31–0.59) |

[a] Adjusted for age and educational attainment

notified with metastatic cancer of known origin. People with CUP were more likely to be in aged care, and their pathway to diagnosis was more likely to involve an emergency presentation and the use of less invasive diagnostic tests. Many ED visits in people with cancer are avoidable;[22] however, the use of less invasive diagnostic tests, for example imaging and fine needle aspiration, may be appropriate for people who are frail (including many aged care residents [23]), or those with a poor prognosis, and this diagnostic pathway is supported by clinical guidelines for CUP [14,21].

This is one of the first studies to identify an increased risk of a CUP diagnosis in aged care residents.[8] Very few cohort studies have examined the association between aged care residency and cancer diagnosis as this information is not typically available at the population-level. A US cancer-registry study of women insured by Medicaid observed a 2.5-fold excess risk of late breast cancer diagnosis in women who were nursing home residents or were in a long-term care facility.[24] A more recent US study[25] found that very few Medicaid-insured nursing home patients received cancer services, and they exhibited a high prevalence of late or unstaged common cancers. While these studies may not be generalizable to the Australian context, it is known that on average, aged care residents are likely to be frail with potentially complex multimorbidity and health care needs. In addition, they may not have close family members to advocate on their behalf regarding changes in their health status. Frail patients may not tolerate cancer treatment due to the increased risk of toxicity and mortality [26], decreasing the need for comprehensive testing to identify the tissue of origin. Late diagnosis may not necessarily be a marker of inadequate care in this setting, as it is possible that functional impairment, care dependency, prognosis, and patient preferences were taken into account when considering the most appropriate diagnostic management of people with metastatic cancer of unknown origin, in keeping with clinical guidelines[13,21].

Confirming previous studies, we found that people with a cancer-registry CUP diagnosis were nearly twice as likely to have had an ED visit in the time period immediately preceding diagnosis, compared to people diagnosed with metastatic cancer of known primary. In our study, 53% of people diagnosed with CUP had a recent ED visit, which is similar to previous Australian studies (50–57%)[8,27] and a UK study of a national cancer registry (57%).[28] Cancer diagnosis via ED presentation is considered a marker for late or delayed diagnosis, and is generally associated with older age, greater deprivation, less access to health services, and poorer prognosis.[29–31] However, the reasons for presentation via ED are complex and many patients also have significant interaction with the health care system prior to their ED presentation, which may represent lost opportunities for diagnosis in individuals with good quality of life and life expectancy who are most likely to benefit from treatment.[31]

While the reasons for ED presentation in people diagnosed with CUP are not known to us, the non-specific and variable nature of CUP symptoms may lead to delayed diagnosis, even in patients who regularly interact with the health care system. As such, we saw no difference between CUP and metastatic cancer of known origin in terms of household income, rural residential location, hospitalisations, GP visits, or allied health visits in this cohort. Although this replicates our previous findings in a veterans cohort,[8] we acknowledge that both studies had limited statistical power, and suggest these associations deserve further scrutiny in large, well-annotated cohorts. It is important to understand the reasons for ED presentation prior to diagnosis, and whether there were missed opportunities for earlier diagnosis, specifically through earlier presentation to a GP, and identification of symptoms suggestive of cancer, leading to suitable investigations.

Only a small proportion of cases and controls had no cancer-related investigations (6%); however, the types of investigations between the two groups differed. People diagnosed with CUP had a greater use of less invasive tests such as serum tumor markers and cytology, and less use of histopathology, compared to controls. While the former are less invasive, the routine use of non-specific serum tumor markers is not recommended for patients with suspected cancer because they can be overexpressed in some people without cancer.[13,32] While histopathology is considered part of the diagnostic approach to identify the primary tumor in CUP patients,[32] only 50% of registered CUP cases in our study were histopathologically verified. This is lower than observed in other population-based studies of CUP[2,7,11,15,33] but higher than our previous study in Australian veterans (36%).[8,15] CUP is a heterogeneous diagnosis, and encompasses individuals who may have undergone exhaustive investigation but a primary site cannot be identified (i.e. people with "true CUP"), and also individuals with a clinical diagnosis only, for whom invasive tests are neither warranted nor desired.[13] In a 2019 US study, only 35% of elderly people with CUP received guideline recommended diagnostic evaluation.[33] Patients with a clinical diagnosis only tend to be older, and have poorer outcomes.[7,8] Unfortunately, we are unable to determine whether the diagnostic approach was clinically appropriate; however, it is recommended that investigations considered to have no impact on prognosis should be avoided.

The strength of this study is the comprehensiveness and quality of the data for a large, prospective cohort; incident cancers and deaths were ascertained by high-quality population-based registries, and we had near complete capture of all subsidized health service interactions for our population, including community-based consultations and tertiary care, diagnostic tests and procedures, as well as information on aged care. This allowed us to construct a comprehensive picture of people's health service use prior to cancer diagnosis. While our sample size is one of the largest population-based studies of people with CUP[8,9], it is modest. As a result, we may have had insufficient power to detect an association with some health services, and where we did, the confidence intervals are wide. While 45 and Up Study participants are healthier on average than the general population, relative estimates calculated from within-cohort comparisons are valid.[34,35] Two data issues are also unlikely to have affected our within-cohort comparisons; firstly, the ED data mainly captured visits to metropolitan and large regional public hospitals (72% and 88% of ED presentations in NSW, respectively).[36,37] Secondly, some pathology services will be under-ascertained as MBS claims data only capture the three most expensive pathology items in an episode of care performed by a GP.[38]

People diagnosed with CUP represent people for whom the primary site could not be identified despite exhaustive testing, as well as those who received minimal diagnostic evaluation; unfortunately we could not distinguish these two groups based on the cancer registry data alone. Ideally, such distinct groups would receive distinguishable diagnoses, but this is not the case. Given the improvement in diagnostics over time, it is likely that the prevalence of "true

CUP" will decrease, with most people diagnosed with CUP representing those who did not receive all clinically-indicated tests, and it is important that these people are not ignored in cancer research. Finally, we have made the assumption that the identified procedures and tests were diagnostic for cancer, but some procedures, particularly non-specific tests such as endoscopy and ultrasound, may have been performed for other reasons. Importantly, we could not directly measure frailty in our study. Frailty plays a role in health decisions and outcome, yet is a concept that is difficult to capture in administrative claims data; while many studies have tried they each have their limitations.[39,40] Our findings support the need for further research to elucidate the relationship between frailty, health service utilisation, and CUP diagnosis.

## Conclusions

We have shown that people with a cancer-registry notified CUP diagnosis had a different pathway to diagnosis than patients with metastatic cancer of known primary. People diagnosed with CUP were more likely to be aged care residents, to have had an ED presentation immediately prior to diagnosis, and to have received fewer invasive tests. Interestingly, despite CUP patients having poorer self-reported overall health,[18] the two groups exhibited no difference in the likelihood of having at least one hospitalisation or consultation during the month of diagnosis and three months prior. Further research is required to examine whether these results indicate missed opportunities in CUP patients for earlier detection and diagnostic testing to reveal the tissue of origin, or conversely represent clinically appropriate differences in management based on patients' underlying health and life expectancy.

## Acknowledgments

We thank the 45 and Up Study, the Department of Human Services (DHS), the NSW Ministry of Health and the NSW Cancer Registry for providing the data used for this study. This research was completed using data collected through the 45 and Up Study (www.saxinstitute. org.au). The 45 and Up Study is managed by the Sax Institute in collaboration with major partner Cancer Council NSW; and partners: the National Heart Foundation of Australia (NSW Division); NSW Ministry of Health; NSW Government Family & Community Services—Ageing, Carers and the Disability Council NSW; and the Australian Red Cross Blood Service. We thank the many thousands of people participating in the 45 and Up Study.

## Author Contributions

**Conceptualization:** Sallie-Anne Pearson, Oscar Perez-Concha, Timothy Dobbins, Robyn L. Ward, Claire M. Vajdic.

**Data curation:** Oscar Perez-Concha.

**Formal analysis:** Oscar Perez-Concha.

**Funding acquisition:** Claire M. Vajdic.

**Methodology:** Andrea L. Schaffer, Sallie-Anne Pearson, Oscar Perez-Concha, Timothy Dobbins, Robyn L. Ward, Marina T. van Leeuwen, Joel J. Rhee, Maarit A. Laaksonen, Glynis Craigen, Claire M. Vajdic.

**Project administration:** Claire M. Vajdic.

**Software:** Oscar Perez-Concha.

**Supervision:** Claire M. Vajdic.

**Visualization:** Oscar Perez-Concha.

**Writing – original draft:** Andrea L. Schaffer.

**Writing – review & editing:** Andrea L. Schaffer, Sallie-Anne Pearson, Oscar Perez-Concha, Timothy Dobbins, Robyn L. Ward, Marina T. van Leeuwen, Joel J. Rhee, Maarit A. Laaksonen, Glynis Craigen, Claire M. Vajdic.

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
