## [Decision Letter · Decision Letter 0]

6 Jan 2020

PONE-D-19-32621

Diagnostic and health service pathways to diagnosis of cancer-registry notified cancer of unknown primary site (CUP)

PLOS ONE

Dear Dr Schaffer,

Thank you for submitting your manuscript to PLOS ONE. After careful consideration, we feel that it has merit but does not fully meet PLOS ONE’s publication criteria as it currently stands. Therefore, we invite you to submit a revised version of the manuscript that addresses the points raised during the review process.

Please respond point-by-point to each comment raised by the reviewers. In terms of Reviewer 1's comments, I do not think you need both confidence intervals and p-values. Please choose one or the other.

We would appreciate receiving your revised manuscript by Feb 20 2020 11:59PM. To enhance the reproducibility of your results, we recommend that if applicable you deposit your laboratory protocols in protocols.io, where a protocol can be assigned its own identifier (DOI) such that it can be cited independently in the future. For instructions see: http://journals.plos.org/plosone/s/submission-guidelines#loc-laboratory-protocols

We look forward to receiving your revised manuscript.

Kind regards,

Erin Bowles

Academic Editor

PLOS ONE

Journal Requirements:

3. Please provide additional details regarding participant consent. In the ethics statement in the Methods and online submission information, please ensure that you have specified (1) whether consent was suitably informed and (2) what type you obtained (for instance, written or verbal). If your study included minors under age 18, state whether you obtained consent from parents or guardians. If the need for consent was waived by the ethics committee, please include this information.

4. We noticed you have some minor occurrence(s) of overlapping text with the following previous publication(s), which needs to be addressed:

https://doi.org/10.1016/j.canep.2019.05.001

https://doi.org/10.1016/j.canep.2019.04.004

In your revision ensure you cite all your sources (including your own works), and quote or rephrase any duplicated text outside the Methods section. Further consideration is dependent on these concerns being addressed.

Reviewers' comments:

Reviewer's Responses to Questions

**Comments to the Author**

1. Is the manuscript technically sound, and do the data support the conclusions?

Reviewer #1: Yes

Reviewer #2: Yes

2. Has the statistical analysis been performed appropriately and rigorously? 

Reviewer #1: Yes

Reviewer #2: Yes

3. Have the authors made all data underlying the findings in their manuscript fully available?

Reviewer #1: Yes

Reviewer #2: Yes

4. Is the manuscript presented in an intelligible fashion and written in standard English?

Reviewer #1: Yes

Reviewer #2: Yes

5. Review Comments to the Author

Reviewer #1: The authors described their methods well and interpretation was guided by the data presented. I do suggest including p-values in tables 1 and 2 as inclusion of variables in the multivariable regression model was based on the p-value. The authors correct in presenting the 95% CI along with the ORs in the narrative but it would be helpful to see the p-values in the data tables. Overall, the paper is well written and the authors explained the rationale of the study well.

Reviewer #2: This paper is an important study examining how patients are diagnosed with cancer of unknown primary (CUP) compared to patients diagnosed with known solid tumors. This knowledge solidly adds to the literature base of CUP. Identification of ED visits confirmed previous VA findings in the general population but also identified aged care residents receipt of CUP diagnoses- care that may be appropriate. Additionally, the authors identify missed opportunity areas to reduce late diagnoses.

Line 79, pg 4, typo: ‘enrolment.’

18% response rate overall. Do you know what percentage of cancer registry persons responded, was it similar to the overall response rate? It seems the use of the survey results was minimal – do you get the same/similar results when not limited to cancer registry-survey patients?

I may have missed it but were the patients with solid tumors limited to late-stage also (stage III or IV)?

Can you clarify what is meant by “GP consultations in the doctors’ rooms” and how that is different in GP consultations in the institution or hospital - I assume it is a doctor's office visit and that is not captured by the institution category.

Pg 7 lines 175, it would be interesting to note the patients who had a clinical diagnosis It seems the results are not mutually exclusive categories (50% had histo, 49% cytological, and 42.2% clinical). The US SEER has about a third of CUP patients who have undergone guideline-recommended diagnostic tests- however, the US is not comparable to Australia because of the US socioeconomic disparities in access to healthcare. I'm curious to know if Australia does a better job. You have described the extent of diagnostic inquiry in more detail further in the manuscript, can you synthesize the findings?

Please clarify the reporting of carcinoma – does that mean that a more detailed assessment was not completed? Does that correlate with patients who only got cytology or clinical diagnosis? I'm assuming this would be comparable to a “not otherwise specified category,” suggesting incomplete evaluation.

Table 2; at first glance, I was confused about ‘<5^3’; took me two readings to figure that out; possibly identify a way to clarify like using parentheses? Also, type on footnote 3 (reaso).

I’d be curious to know if aged care residents are more likely to visit ED than non-aged care patients and if this was driving the ED finding. Does interaction/effect modification play a role here? The sample size may to too small to assess this, just curious.

6. PLOS authors have the option to publish the peer review history of their article (what does this mean?). If published, this will include your full peer review and any attached files.

Reviewer #1: No

Reviewer #2: Yes: Julie Smith-Gagen

---

## [Author Response · Author response to Decision Letter 0]

19 Feb 2020

February 19, 2020

Dear Dr Bowles,

Thank you for the opportunity to revise and resubmit our manuscript, “Diagnostic and health service pathways to diagnosis of cancer-registry notified cancer of unknown primary site (CUP)”. We have ensured that we have cited all previous works using the same cohort, and that there is no overlapping text outside the Methods section.

The data used for this study are from the 45 and Up Study, which is an open research resource managed by the Sax Institute (https://www.saxinstitute.org.au/our-work/45-up-study/for-researchers/). The datasets for this study were created by linkage of the 45 and Up Study baseline survey data to Australian Government and NSW state data sources with support from the NSW Centre for Health Record Linkage (www.cherel.org.au), and permission from the custodians of the datasets under specific ethics approvals for pre-specified analyses only. These data contain sensitive information and are potentially re-identifiable, and we do not have permission to share them with third parties. The data used for this study are available to researchers pending appropriate ethics and data access requirements. Interested researchers can contact the Sax Institute (45andup.research@saxinstitute.org.au) and NSW Centre for Health Record Linkage (cherel.mail@moh.health.nsw.gov.au) for data access approval procedures.

We have also included further information about consent and ethics in the Methods section of the manuscript on Page 7.

We have addressed all reviewers’ comments as described below. A formatted version of our response is also included in the attached cover letter as a pdf.

On behalf of all co-authors,

Andrea Schaffer, MSc, MBiostat, PhD

Centre for Big Data Research in Health

Level 2, AGSM Building (G27)

UNSW Sydney

Sydney AUSTRALIA 

******

Reviewer #1: The authors described their methods well and interpretation was guided by the data presented. I do suggest including p-values in tables 1 and 2 as inclusion of variables in the multivariable regression model was based on the p-value. The authors correct in presenting the 95% CI along with the ORs in the narrative but it would be helpful to see the p-values in the data tables. Overall, the paper is well written and the authors explained the rationale of the study well.

RESPONSE: Thank you to the reviewer for their positive comments. The p-value was only one consideration for choosing which covariates to include in the multivariable model, as we considered correlation between covariates and clinical expertise to avoid including variables that were on the same pathway to a CUP diagnosis. We believe that there is limited utility in including p-values in addition to confidence intervals. As advised by the Editors we have chosen to present confidence intervals only. 

*******

Reviewer #2: This paper is an important study examining how patients are diagnosed with cancer of unknown primary (CUP) compared to patients diagnosed with known solid tumors. This knowledge solidly adds to the literature base of CUP. Identification of ED visits confirmed previous VA findings in the general population but also identified aged care residents receipt of CUP diagnoses- care that may be appropriate. Additionally, the authors identify missed opportunity areas to reduce late diagnoses.

RESPONSE: Thank you to the reviewer for her comments. 

Line 79, pg 4, typo: ‘enrolment.’

RESPONSE: “Enrolment” is the standard spelling in British and Australian English. However, given that PLOS ONE is published by an American organisation we have now used standard American English spellings, including “enrollment”, throughout the manuscript.

18% response rate overall. Do you know what percentage of cancer registry persons responded, was it similar to the overall response rate? 

RESPONSE: We do not know the response rate for people with a history of cancer and registered cancer. The decision to participate in the 45 and Up Study and completion of the baseline questionnaire occurred prior to the registry-notified cancer diagnoses in this study. 

It seems the use of the survey results was minimal – do you get the same/similar results when not limited to cancer registry-survey patients?

RESPONSE: We did evaluate all survey results and found that information from the survey was minimally predictive of a CUP diagnosis (except for educational attainment) after including health service utilisation in our model. However, this may be because the median time between completion of the baseline questionnaire and the CUP diagnosis was 33 months. Further, while we had information on self-rated health and quality of life, they likely are measuring similar constructs as residence in aged care (which was the strongest predictor of a CUP diagnosis), specifically frailty and poorer health status. 

We performed a similar study in Australian veterans (Vajdic et al. Health service utilisation and investigations before diagnosis of cancer of unknown primary (CUP): A population-based nested case-control study in Australian Government Department of Veterans’ Affairs clients. Cancer Epidemiol 2015;39:585-592) and also found that people with a registry-notified diagnosis of CUP were more likely to be in aged care compared with people with metastatic cancer of known primary. We have cited this study in the Discussion when discussing the findings from our current study. 

I may have missed it but were the patients with solid tumors limited to late-stage also (stage III or IV)?

RESPONSE: Controls with known primary solid tumours were eligible if the first manifestation of their cancer was metastatic, with either regional or distant metastases, as classified by the NSW Cancer Registry, as described on Page 5. Note that we allowed regional metastases in the control group as there exist “true CUP” cases with regional metastases, e.g. neck node for presumed head and neck primary; axillary lymph node for presumed breast cancer primary. We did not have detailed information on cancer stage in our data. 

Can you clarify what is meant by “GP consultations in the doctors’ rooms” and how that is different in GP consultations in the institution or hospital - I assume it is a doctor's office visit and that is not captured by the institution category.

RESPONSE: The reviewer is correct, GP consultations in doctors’ rooms is in the doctor’s office; the former is Australian terminology and is equivalent to “doctor’s office”. We have separated this measure from GP consultations in a home, institution or hospital as we expected that the latter would be a potential marker for frailty or poorer health, given that they represent visits to patients who are hospitalised, in aged care, or who cannot travel to the doctor’s office. We have now clarified in the Methods that “doctors’ rooms” is the same as “doctor’s office” (Page 6).

Pg 7 lines 175, it would be interesting to note the patients who had a clinical diagnosis It seems the results are not mutually exclusive categories (50% had histo, 49% cytological, and 42.2% clinical). The US SEER has about a third of CUP patients who have undergone guideline-recommended diagnostic tests- however, the US is not comparable to Australia because of the US socioeconomic disparities in access to healthcare. I'm curious to know if Australia does a better job. You have described the extent of diagnostic inquiry in more detail further in the manuscript, can you synthesize the findings?

RESPONSE: Thank you for noticing this error, as the categories should in fact be mutually exclusive. In the text we now describe the highest level of diagnostic enquiry: 

Page 7: “Of the people with a registry-notified CUP diagnosis, 165 (50.5%) had a histopathogical diagnosis, 49 (15.0%) a cytological diagnosis (including fine needle aspiration, smears, washing and sputum), 88 (26.9%) a clinical diagnosis (including clinical, imaging and biochemical procedures) and 25 (7.6%) were identified by death certificate only."

There are no Australian guidelines for diagnosis of CUP. However, we have now included data on the number of people who did not have any cancer-related diagnostic procedures (i.e. imaging, surgery, pathology tests) in both the CUP cases and controls, and included this information in the Results:

Page 8: “Overall, 21 (6.4%) people diagnosed with CUP did not have any cancer-related investigations, as defined by imaging procedures, endoscopy, medical procedures (e.g. resection), or pathology tests. Among controls, 60 (6.1%) did not have any cancer-related investigations.”

We have also included the following sentence in the Discussion, regarding the disparity in types of procedures that the two groups underwent:

Page 14: “Only a small proportion of cases and controls had no cancer-related investigations (6%); however, the types of investigations between the two groups differed. People diagnosed with CUP had a greater use of less invasive tests such as serum tumor markers and cytology, and less use of histopathology, compared with controls.”

Lastly, we were not aware of the mentioned study at the time of submission of our manuscript; it is highly relevant and we have now cited it in the Discussion. 

Please clarify the reporting of carcinoma – does that mean that a more detailed assessment was not completed? Does that correlate with patients who only got cytology or clinical diagnosis? I'm assuming this would be comparable to a “not otherwise specified category,” suggesting incomplete evaluation.

RESPONSE: Of the 165 people with a histopathological diagnosis, 101 (61%) were classified as carcinoma. Of these, 20 were carcinoma not otherwise specified and 81 were a specified carcinoma subtype. We are unable to report specific carcinoma subtypes because most subtypes have less than 5 cases and for privacy reasons we cannot report counts <5. We have modified the text to include the above information:

Page 8: “Of the CUP cases (n=165) registered with a NSW Cancer Registry histopathological diagnosis, 20 (12.1%) tumors were carcinoma (subtype not specified), 81 (49.1%) were carcinoma (subtype specified), 50 (30.3%) were adenocarcinoma, and 14 (8.5%) were other morphological types.”

Table 2; at first glance, I was confused about ‘<5^3’; took me two readings to figure that out; possibly identify a way to clarify like using parentheses? Also, type on footnote 3 (reaso).

RESPONSE: Thank you, we have corrected the typo. For clarity, we have now replaced all table footnote indicators with letters, rather than numbers, which may be confused with numerical content in the tables. “<5^3” in the cells has now been replaced with the letter “c” and we have included relevant information in the footnote.

I’d be curious to know if aged care residents are more likely to visit ED than non-aged care patients and if this was driving the ED finding. Does interaction/effect modification play a role here? The sample size may to too small to assess this, just curious.

RESPONSE: The increased risk of a CUP diagnosis in people with an ED visit is adjusted for residence in an aged care facility, and thus is unlikely to be driven by people in aged care. There was little correlation between residency in aged care and having had an ED visit (Cramér's V statistic=0.12), suggesting that the rate of ED visits was similar between those in aged care and those not. Lastly, detection of significant interactions requires a much larger sample size than for main effects, and as the reviewer correctly points out our sample size is likely too small to detect most interactions.

---

## [Editor Report · Decision Letter 1]

28 Feb 2020

Diagnostic and health service pathways to diagnosis of cancer-registry notified cancer of unknown primary site (CUP)

PONE-D-19-32621R1

Dear Dr. Schaffer,

We are pleased to inform you that your manuscript has been judged scientifically suitable for publication and will be formally accepted for publication once it complies with all outstanding technical requirements.

With kind regards,

Erin Bowles

Academic Editor

PLOS ONE
---

## [Editor Report · Acceptance letter]

4 Mar 2020

PONE-D-19-32621R1 

Diagnostic and health service pathways to diagnosis of cancer-registry notified cancer of unknown primary site (CUP) 

Dear Dr. Schaffer:

I am pleased to inform you that your manuscript has been deemed suitable for publication in PLOS ONE. Congratulations! Your manuscript is now with our production department. 

With kind regards,

on behalf of

Dr. Erin Bowles 

Academic Editor

PLOS ONE